# Automatic Transcription for Estonian Children's Speech

**Agnes Luhtaru**[α]   **Rauno Jaaska**[α]   **Karl Kruusamäe**[τ]   **Mark Fishel**[α]

[α]: Institute of Computer Science, University of Tartu, Estonia
[τ]: Institute of Technology, University of Tartu, Estonia

`{agnes.luhtaru, rauno.jaaska, karl.kruusamae, mark.fisel}@ut.ee`

## Abstract

We evaluate the impact of recent improvements in Automatic Speech Recognition (ASR) on transcribing Estonian children's speech. Our research focuses on fine-tuning large ASR models with a 10-hour Estonian children's speech dataset to create accurate transcriptions. Our results show that large pre-trained models hold great potential when fine-tuned first with a more substantial Estonian adult speech corpus and then further trained with children's speech.

## 1   Introduction

Automatic Speech Recognition (ASR) continues to face challenges in accurately transcribing children's speech. Research efforts are underway to adapt adult ASR models to better handle the unique pronunciation variations and limited vocabulary that are characteristic of children's speech (Thienpondt and Demuynck, 2022; Dutta et al., 2022). These adaptations are necessary due to the limitations of current ASR systems, which often lack adequate representation of children's speech and struggle to generalize to new examples.

Recent advancements in ASR technology, including the use of large transformer-based models and unsupervised pre-training techniques, have resulted in improved performance for adult speech recognition, with the ability to train on a diverse range of data without human annotations (Baevski et al., 2020; Radford et al., 2022; Hsu et al., 2021). These models demonstrate greater robustness and generalization compared to previous systems. However, the effectiveness of these advanced ASR models for children's speech, especially in low-resource languages like Estonian, remains untested.

In this paper, we are investigating two multilingual speech models - Facebook's Wav2Vec2-XLS-R (Babu et al., 2021) and OpenAI's Whisper (Radford et al., 2022) - as potential starting points for building an ASR system transcribing Estonian children's speech. Our objective is to determine the potential of these models in creating low-effort ASR systems for children speaking a low-resource language like Estonian, for which there are no ASR systems for children's speech.

To accomplish this, we fine-tune the XLS-R and Whisper models from scratch using children's speech data. We also fine-tune pre-existing models for the Estonian language with additional children's speech recordings. Furthermore, we compare the quality of the ASR system by evaluating a pre-made Estonian ASR system provided by Microsoft Azure and exploring its fine-tuning capabilities.

Our research indicates that XLS-R models and Whisper models can serve as effective starting points for building an ASR system using only 10 hours of children's speech. However, for optimal performance, these models should first be fine-tuned with Estonian adult speech. We achieve the best word error rate of around 15 using an XLS-R model that was fine-tuned with Estonian ASR datasets and further trained with children's speech. Furthermore, our results show that the Azure speech-to-text model performs similarly to the Estonian XLS-R and Whisper models but not as well as the fine-tuned public models. Two models that achieved the lowest WER scores are available in HuggingFace[1][2].

In the next sections, we describe which data we used for evaluation and training, which models we used and how we fine-tuned these and last but not

---

[1]`https://huggingface.co/tartuNLP/xls-r-300m-et-children`
[2]`https://huggingface.co/tartuNLP/whisper-large-v2-et-children`

least we present and analyse the results.

## 2 Dataset and evaluation

The Children ASR dataset used in this work consists of speech recordings from 53 children aged 6 to 13. The data was collected by the Children's Clinic of Tartu University Hospital and contains a mix of both boys and girls speaking about various topics such as answering questions, describing pictures, talking about their family and friends, and more. The dataset is divided into three subsets - test, dev, and train - with no overlap in speakers or texts.

The test set contains all age and gender groups and has a total recording duration of 278 minutes (approximately 4.6 hours). The development set is missing some speakers and has a total recording duration of 182 minutes (approximately 3 hours). The training set is also missing some speakers and has a total recording duration of 613 minutes (approximately 10 hours). A breakdown of the total recording duration for the test set by age and gender of the speakers is shown in Table 1.

| Age | Girls (min) | Boys (min) | Total (min) |
|---|---|---|---|
| 6 | 17 | 21 | 38 |
| 7 | 14 | 16 | 30 |
| 8 | 17 | 14 | 31 |
| 9 | 22 | 18 | 40 |
| 10 | 15 | 17 | 32 |
| 11 | 20 | 17 | 37 |
| 12 | 16 | 22 | 38 |
| 13 | 19 | 13 | 32 |
| Total | 140 | 138 | 278 |

Table 1: Total recording duration in minutes for the Estonian children ASR test set, broken down by age and gender of the speakers.

The children in the dataset speak about a wide range of topics, covering everything from answering questions and describing pictures to discussing their family and friends. They also include recordings of children reading fairytales, reciting poems, and saying specific sentences. The utterances in the dataset vary in their level of spontaneity - some are unscripted expressions of thoughts, while others feature children reading.

We evaluate the performance of our speech recognition models using the standard measure of word error rate (WER). This involves converting all text to lowercase and removing punctuation but not standardizing different spelling variations. Our reference transcriptions reflect the pronunciation of children, including any errors they may make. However, the line between correct and incorrect pronunciation is often blurry and some children's speech can be difficult to comprehend. We do not consider the ambiguity in human transcriptions and simply compare the models' output to our reference transcription, which could lead to increased WERs.

## 3 Models and training

We are using both public large speech models and private black box speech service. In the case of public models, we also searched for models already fine-tuned with Estonian speech data. We fine-tune the selection of these models with the children's speech dataset mentioned in the last section.

For public models, we use two multilingual ones: Facebook's XLS-R and OpenAI's Whisper (Radford et al., 2022). XLS-R model is trained with speech modelling objective, not ASR but it can be fine-tuned to ASR with Connectionist Temporal Classification (CTC) (Graves et al., 2006) algorithm. The Whisper on the other hand is a multipurpose model that contains both transformer encoder and decoder blocks and has been trained on several speech-processing tasks, like multilingual speech recognition, speech translation and voice activity detection (Radford et al., 2022).

The available XLS-R models have 300 million, 1 billion and 2 billion parameters, we are using the two smaller ones in this work. The Whisper model comes in six different sizes; we are using medium and large-v2 since the Estonian error rates for other ones are relatively high. There is one Estonian-specific fine-tuned model available for the 300 million parameter version, trained with over 700 hours of Estonian speech data (Alumäe and Olev, 2022). There are several Estonian Whisper models available in HuggingFace but these are trained with fewer data examples. We are using the best available medium and large-v2 ones.[3][4]. Following the submission of this paper, a new Estonian Whisper model was released[5], which is

---

[3] https://huggingface.co/agnesluhtaru/whisper-medium-et-ERR2020
[4] https://huggingface.co/agnesluhtaru/whisper-large-et-ERR2020-v2
[5] https://huggingface.co/TalTechNLP/whisper-medium-et

trained using a larger dataset. In the scope of this work, we evaluate the model but do not fine-tune it using children's speech.

We use standard fine-tuning procedures. For training XLS-R-based ASR models from scratch, we use the learning rate of 3e-4, a 400-step warmup and train the models for 60 epochs with children's speech dataset, which is less than 4000 steps. When further fine-tuning the Estonian XLS-R model with children's speech, we use the learning rate of 2e-5 and 200 warmup steps. We fine-tune all the Whisper models with warmup 10% of the steps and learning rate 1e-05. When fine-tuning the out-of-the-box Whisper models, we train these for 5000 steps or atound 40 epochs and when fine-tuned models already trained with Estonian adult speech, we train the large model for 2000 steps or over 16 epochs and medium model for 1000 steps or eight epochs.

For the private model, we use Microsoft Azure Speech service's speech-to-text[6], which requires an Azure subscription and a Speech resource. The transcription services can be accessed by making REST requests.

Microsoft Azure offers the option to fine-tune the model with custom datasets. This process involves uploading data to train the models, followed by deploying the trained models. Since audio-based fine-tuning is not available for Estonian, we use text-based tuning for our work with the texts from the children's speech dataset.

# 4 Results

In this section, we describe the results of all the models based on Facebook's XLS-R, OpenAI'S Whisper and Microsoft Azure speech-to-text.

## 4.1 XLS-R

Table 2 shows the word error rate (WER) scores of fine-tuned Estonian XLS-R models using only 10 hours of Estonian children's speech data, the fine-tuned Estonian model (Alumäe and Olev, 2022) and Estonian model further trained with children's speech. We can see that the limited amount of data for fine-tuning XLS-R from scratch results in a high WER of over 30 for both models with 300 million and one billion parameters. Training an ASR model using only 10 hours of speech data

[6]https://learn.microsoft.com/en-us/azure/cognitive-services/speech-service/speech-to-text

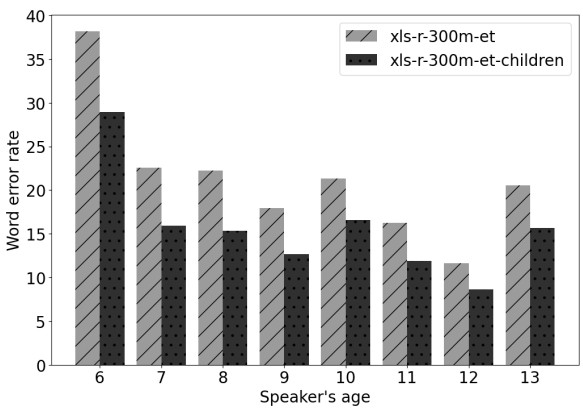

Figure 1: Performance comparison of Estonian XLS-R ASR and children's speech fine-tuned models across age groups.

can be challenging, especially when the speech is for a low-resource language and children.

| Model | Dev | Test |
|---|---|---|
| xls-r-300M-children | 34.58 | 36.3 |
| xls-r-1B-children | 31.06 | 30.89 |
| xls-r-300M-et | 19.15 | 20.62 |
| xls-r-300M-et-children | 14.30 | 15.31 |

Table 2: Comparison of WER scores for Facebook's Wav2Vec2 XLS-R (Babu et al., 2021) based models fine-tuned with only Estonian children's speech, only Estonian adult speech (Alumäe and Olev, 2022) and first fine-tuned to Estonian and further trained with children's speech.

The results show that the pre-trained Estonian ASR model has a WER of around 20, while further fine-tuning the model with children's speech data leads to even better results, with a WER of less than 15. Based on the lower WER score for fine-tuned one billion parameter model, we can suggest that a larger model fine-tuned with Estonian data first and then further trained on children's speech could lead to even better results.

The results indicate that fine-tuning the Estonian ASR model using children's speech data improves performance across all age groups (refer to Figure 1). Younger speakers tend to have a higher word error rate (WER) than older speakers, although this relationship is not always straightforward. There are some exceptions, such as the recognition performance for 13-year-olds being worse than that of younger age groups. This high-

lights that speaker variability plays a role in the WER results. Nevertheless, the fine-tuning of the ASR model using children's speech data reduces the differences in recognition performance across age groups, resulting in improved overall performance.

## 4.2 Whisper

The performance of the out-of-the-box Whisper models on the children's dataset (see Table 3) is comparable to the scores reported by Radford et al. (2022) on the Estonian Common Voice 9 Ardila et al. (2020). All models have a WER of at least 35. So, although we can use Whisper without fine-tuning, it does not transcribe Estonian speech well and therefore does not give great transcriptions for Estonian children's speech as well.

When fine-tuning the model using only 10 hours of children's speech, we can already achieve better results. The large-v2 model yields a WER of around 20, which is significantly better than some models fine-tuned with Estonian speech alone. The medium model, developed by Tal-TechNLP and trained with over 800 hours of Estonian speech[7], outperforms the XLS-R model that was trained solely on Estonian adult speech.

| Model | Dev | Test |
|---|---|---|
| Whisper-medium | 43.21 | 46.11 |
| Whisper-large-v2 | 35.06 | 36.01 |
| Whisper-medium-children | 24.29 | 25.08 |
| Whisper-large-v2-children | 20.58 | 20.38 |
| TalTech Whisper-medium-et | 15.64 | 17.26 |
| Whisper-medium-et | 26.83 | 28.78 |
| Whisper-large-v2-et | 28.13 | 29.2 |
| Whisper-medium-et-children | 17.49 | 18.66 |
| Whisper-large-v2-et-children | 15.73 | 16.02 |

Table 3: Comparison of WER scores for OpenAI Whisper (Radford et al., 2022) models and Whisper models fine-tuned with only Estonian children's speech, only Estonian adult speech and first fine-tuned to Estonian and further trained with children's speech.

Despite using the Estonian Whisper models fine-tuned with fewer audio text pairs than the XLS-R model, when trained further with children's speech, the large model achieved similar WER as the double fine-tuned

---

[7]https://huggingface.co/TalTechNLP/whisper-medium-et

smaller XLS-R model. The difference between TalTechNLP's whisper-medium-et and whisper-large-v2-et-children is small, suggesting that fine-tuning the former with children's data could potentially result in even better performance.

## 4.3 Azure

The results from our evaluation of the children's speech dataset show that the out-of-the-box Azure speech-to-text model performs similarly or better than the fine-tuned Estonian XLS-R model (Alumäe and Olev, 2022) but worse than Estonian Whisper medium trained by TalTechNLP. As indicated in Table 4, the Microsoft Azure speech-to-text scores are around 20 or below.

| Model | Dev | Test |
|---|---|---|
| Microsoft Azure | 20.18 | 18.93 |
| Azure text-tuned | 21.21 | 20.31 |

Table 4: WER scores for Microsoft Azure speech-to text and its custom text-tuned version.

However, the experiment also shows that text-tuning is not the best approach for this particular dataset. The dataset mostly contains simpler vocabulary and not much terminology, most likely leading to quick overfitting with text-tuning. Currently, text-tuning is the only option available for the Estonian language, but it might not be the best use case for children's speech datasets.

## 5 Discussion

Our experiments show that children's speech recognition continues to be a tricky problem but big speech models are looking promising. It is possible to build an ASR system for Estonian children's speech without any bells and whistles using only 10 hours of data and get output that is decent and might be good enough for use in chatbots. However, when it comes to six-year-olds, whose speech is difficult to understand even for the human ear, the system is still struggling.

We evaluate different models and it appears that both OpenAI's Whisper and Facebook's XLS-R are viable options for developing a speech recognition model for Estonian children's speech. The current best word error rate is around 15 with XLS-R. However, it remains unclear if this pretrained model is optimal for children's speech or if a lower error rate could be achieved with Whisper after fine-tuning with a similar amount of Estonian

adult speech. Additionally, we do not obtain comparable results with the Azure service, as it does not permit fine-tuning with audio data.

Our findings suggest that the results could be improved by using a larger XLS-R model as the base or by fine-tuning Whisper models with more data. Additionally, we do not use a separate language model, which is possible with both Whisper and XLS-R models and could potentially enhance the performance of these models.

# 6 Conclusion

We test the performance of two speech recognition models, XLS-R and Whisper, on transcribing Estonian children's speech. We fine-tune the models with children's speech data and compared them to an off-the-shelf system from Microsoft Azure. Both models fine-tuned with children's speech, outperform Microsoft Azure, which does not allow fine-tuning with audio for Estonian, and are promising for children's ASR system.

# Acknowledgements

This research has been in part supported by European Social Fund via IT Academy programme, Estonian Centre of Excellence in IT (EXCITE) funded by the European Regional Development Fund, and AI & Robotics Estonia co-funded by the EU and Ministry of Economic Affairs and Communications in Estonia.

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
