# OpenReview forum: "Automatic Transcription for Estonian Children’s Speech"
_NoDaLiDa/2023/Conference — NoDaLiDa 2023_

### Official Review · Reviewer_mnCe · 2023-03-09
**A comparison between various end-to-end ASR models for low-resource children's speech recognition**

**Rating:** 8
**Confidence:** 5

**Review:**

The paper introduces a new dataset consisting of approx. 10 hours of Estonian children's speech and demonstrates the effectiveness of two modern systems (wav2vec2 and Whisper) on this data. Furthermore, a comparison is made with the commercial ASR system powered by Microsoft Azure.

Strengths:
* Three different approaches are compared using the same data
* The continued fine-tuning (first train with adult speech then use the children data) approach seems interesting, and the results confirm that it is beneficial.
* Efforts were made to adapt even the Azure ASR system to the data
* The reported WERs are convincing.

Weaknesses:
* The dataset description requires a bit more details, like vocabulary size, average length of recordings, the ratio of data per topic. Additionally, Table 1 should contain the statistics for the training and dev data too.
* The study only uses WER as the only evaluation metric, but character error rates could give the readers a more in-depth information about the performance of the models, especially as the mispronounced words could inflate the WER metric and paint a false picture about the real performance.

Overall, the paper presents a very nice solution for low-resource children's ASR, and could guide others to choose the best model and training approach in similar situations.

Minor comments:
* Only an overall WER is given on the test set, I would be interested in the per-topic/task WERs. Is XLS-R the best for recognizing speech in the case of question answering or picture description?
* Have the authors compare the agreement between systems? Do Whisper and XLS-R struggle with the same recordings/words?
* A hybrid HMM/DNN baseline would give us an idea of the importance of using a self-supervised model.

**Paper Type:**

Short paper

---

### Official Review · Reviewer_iAWh · 2023-03-09
**Automatic Transcription for Estonian Children’s Speech**

**Rating:** 8
**Confidence:** 4

**Review:**

The paper provides a nice overview of efforts to produce ASR for Estonian children’s speech by various finetuning strategies for large pre-trained models, namely wav2vec2 XLS-R and Whisper.

Child speech recognition is an important topic that lacks both data and sufficient scientific and technological insight to reach comparable performance to adult speech recognition.  It is therefore nice to see this research.

The paper is well presented using clear language and a good structure.

While the paper is based on existing systems and architectures, as well as standard fine-tuning techniques, and as such does not present any significant technological advances, it does provide insights on the adaptation to language and age groups that has interest also for other languages.

The systems explored do not employ separate language models, and the paper comments that this “could potentially enhance the performance of these models”. This is absolutely true, and should be explored in further work.

In the description of the data used in this study (Section 2) it is stated that
>Our reference transcriptions reflect the pronunciation of children, including any errors they may make,

I am unsure of how this is to be interpreted. Does it mean that words are transcribed as they are pronounced, which may be an “illegal” spelling? E.g. (English example) if a child mispronounces “dandelion” as “dalin”, the transcription will read “dalin”? Or do errors refer to hesitations, use of wrong words (but “legal” spelling)?


**Paper Type:**

Short paper

---

### Official Review · Reviewer_LA7Y · 2023-03-10
**ASR for Estonian Children's Speech**

**Rating:** 7
**Confidence:** 5

**Review:**

The paper presents evaluation for fine-tuning large pre-trained models (wav2vec2-xls-r and Whisper) to Estonian Children's speech.  The paper shows what it is better to use a larger Estonian adult speech dataset first before fine-tuning with the smaller Estonian children dataset.  The commercial ASR system from Microsoft Azure is also used but can't be fine-tuned using acoustic data, only text (for language modelling).  Unfortunately direct comparison between the xls-r and Whisper is not possible because the fine-tuning is not the same, so the results have to be interpreted separately.  Nevertheless, the paper is a worthy contribution to the conference and should be accepted if possible.

**Paper Type:**

Short paper

---

### Decision · Program_Chairs · 2023-03-17

Accept